# Mediterranean diet effects on vascular health and serum levels of adipokines and ceramides

Mario Daidone[1,2], Alessandra Casuccio[1,2], Maria Grazia Puleo[1,2], Alessandro Del Cuore[1,2], Gaetano Pacinella[1,2], Tiziana Di Chiara[1,2], Domenico Di Raimondo[1,2], Palmira Immordino[1], Antonino Tuttolomondo[1,2]*

1 Dipartimento di Promozione della Salute, Materno-Infantile, di Medicina Interna e Specialistica di Eccellenza "G. D'Alessandro", Palermo, Italy, 2 U.O. C di Medicina Interna con Stroke Care, Palermo, Italy

* bruno.tuttolomondo@unipa.it

## Abstract

### Background

A randomized clinical trial to evaluate the effect of a Mediterranean-style diet on vascular health indices such as endothelial function indices, serum lipid and ceramide plasma and some adipokine serum levels. We recruited all consecutive patients at high risk of cardiovascular diseases admitted to the Internal Medicine and Stroke Care ward at the University Hospital of Palermo between September 2017 and December 2020.

### Materials and methods

The enrolled subjects, after the evaluation of the degree of adherence to a dietary regimen of the Mediterranean-style diet, were randomised to a Mediterranean Diet (group A) assessing the adherence to a Mediterranean-style diet at each follow up visit (every three months) for the entire duration of the study (twelve months) and to a Low-fat diet (group B) with a dietary "counselling" starting every three months for the entire duration of the study (twelve months).The aims of the study were to evaluate: the effects of adherence to Mediterranean Diet on some surrogate markers of vascular damage, such as endothelial function measured by means of the reactive hyperaemia index (RHI) and augmentation index (AIX), at the 6-(T1) and 12-month (T2) follow-ups; the effects of adherence to Mediterranean Diet on the lipidaemic profile and on serum levels of ceramides at T1 and T2 follow-ups; the effects of adherence to Mediterranean Diet on serum levels of visfatin, adiponectin and resistin at the 6- and 12-month follow-ups.

### Results

A total of 101 patients were randomised to a Mediterranean Diet style and 52 control subjects were randomised to a low-fat diet with a dietary "counselling". At the six-month follow-up (T1), subjects in the Mediterranean Diet group showed significantly lower mean serum total cholesterol levels, and significantly higher increase in reactive hyperaemia index (RHI) values compared to the low-fat diet group. Patients in the Mediterranean Diet group also

**Funding:** The author(s) received no specific funding for this work.

**Competing interests:** I have no competing interests

**Abbreviations:** PREDIMED, Prevention with Mediterranean DietCVD: cardiovascular disease; CHD, cardiac heart disease; BMI, Body mass index; ADA, American Diabetes Association; ESC/ESH, European Society of Cardiology/ European Society of Hypertension; NCEP-ATP III, National Cholesterol Education Programme Adult Treatment Panel III; CT, compurized thomography; MRI, magnetic resonance imaging at; F.F.Q.s, semiquantitative food frequency questionnaire; LDL, low density lipoprotein; HDL, high density lipoprotein; RHI, reactive hyperaemia index; AIX, augmentation index; RH-PAT, reactive hyperaemia-amplitude tonometry; C24:0, C24:0 ceramide; C16:0, C16:0 ceramide; C22:0, C22:0 ceramide; C18:0, C18:0 ceramide; C24:0/C16:0, C24:0/C16:0 ratio; ANOVA, Univariate analysis of variance; NO, nitric oxide; IL-12, interleukin-12; IL-6, interleukin6; TNF-α, Tumour necrosis factor; MCP-1, monocyte chemoattractant protein-1; PBMCs, peripheral blood mononuclear cells; NF-κB, nuclear factor-κB; ICAM-1, Intercellular adhesion molecule-1; VCAM-1, vascular-Intercellular adhesion molecule-1.

showed lower serum levels of resistin and visfatin at the six-month follow-up compared to the control group, as well as higher values of adiponectin, lower values of C24:0, higher values of C22:0 and higher values of the C24:0/C16:0 ratio. At the twelve-month follow-up (T2), subjects in the Mediterranean Diet group showed lower serum total cholesterol levels and lower serum LDL cholesterol levels than those in the control group. At the twelve-month follow-up, we also observed a further significant increase in the mean RHI in the Mediterranean Diet group, lower serum levels of resistin and visfatin, lower values of C24:0 and of C:18:0, and higher values of the C24:0/C16:0 ratio.

## Discussion

The findings of our current study offer a further possible explanation with regard to the beneficial effects of a higher degree of adherence to a Mediterranean-style diet on multiple cardiovascular risk factors and the underlying mechanisms of atherosclerosis. Moreover, these findings provide an additional plausible interpretation of the results from observational and cohort studies linking high adherence to a Mediterranean-style diet with lower total mortality and a decrease in cardiovascular events and cardiovascular mortality.

## Trial registration

ClinicalTrials.gov Identifier: NCT04873167. https://classic.clinicaltrials.gov/ct2/show/NCT04873167.

## Background

The Mediterranean Diet, as originally defined by Ancel Keys, is characterized by the dietary habits observed in Mediterranean natives during the 1960s, emphasizing low saturated fat and high vegetable oil intake [1,2]. Over time, our understanding of nutrition has evolved, prompting a shift towards investigating total dietary patterns. The concept of a "dietary scheme" has emerged, involving a comprehensive analysis of food and nutrient consumption within specific dietary patterns [3].

Recent research has underscored the pivotal role of the Mediterranean-style diet in preventing cardiovascular diseases. The Lyon Diet Heart Study demonstrated its positive impact on secondary prevention [4]. The landmark PREDIMED study, a large-scale randomized trial, provided compelling evidence for the effectiveness of the Mediterranean-style diet in primary prevention, significantly reducing cardiovascular events compared to a low-fat diet [5–7].

Cardiovascular diseases, including coronary artery disease, cerebrovascular disease, peripheral artery disease, and heart failure, remain the leading global cause of mortality. Atherosclerosis, the main culprit behind these diseases, has a complex pathogenesis involving genetic, metabolic, environmental, and behavioral factors [8]. Hypercholesterolemia, a major metabolic risk factor, contributes significantly to atherosclerosis. Studies consistently highlight the role of reducing fat in the diet, particularly saturated fat, in lowering plasma cholesterol levels and reducing the incidence of coronary heart disease. Notably, the substitution of saturated fat with polyunsaturated and monounsaturated fats, such as those found in virgin olive oil, has shown positive effects [9,10].

Despite established clinical and biochemical risk predictors, a substantial residual risk persists, necessitating the identification of additional predictive biomarkers. Ceramides,

belonging to the sphingolipid family, have been implicated in cellular dysfunction and insulin resistance. Excessive de novo ceramide biosynthesis, triggered by stimuli like high serum levels of saturated free fatty acids, may contribute to the pathogenesis of cardiometabolic diseases, including insulin resistance and low-grade inflammation [11,12]. Therefore, evaluating serum ceramide levels becomes crucial in understanding the intricate interplay between lipidaemic pathways and vascular health.

Adipocytes, the cells responsible for storing and releasing fat, produce hormones that play a crucial role in the local microenvironment and distant tissues. Adipokines, specific hormones secreted by adipose tissue, have been linked to inflammation, atherosclerosis, and related cardiovascular complications [13].

Despite advancements in understanding, few studies have prospectively analyzed the association between ceramides and the incidence of cardiovascular and cerebrovascular events. Laaksonen et al. reported an unclear relationship between plasma ceramides and death from cardiovascular disease [14]. Additionally, the PREDIMED study suggested that dietary interventions following a Mediterranean-style diet positively modified the association between a blood ceramide score (comprising individual ceramides and ceramide ratios) and the risk of incident CVD [15]. The CORDIOPREV study in patients with established coronary heart disease concluded that the Mediterranean diet was superior to the low-fat diet in preventing major cardiovascular events in secondary prevention [16].

Despite these findings, no study has comprehensively evaluated the effects of the Mediterranean Diet on endothelial function, in parallel with its impact on serum lipid changes, including ceramide changes, and on specific adipokine pathways. In response to this gap, we have designed a randomized trial to systematically investigate the effects of the Mediterranean diet on endothelial function, measured by the evaluation of the reactive hyperaemia index (RHI), as well as its impact on serum ceramide and adipokine levels. This research aims to provide a holistic understanding of the intricate connections between diet, vascular health, and cardiometabolic risk factors.

## Study hypothesis

The study hypothesis of our randomized parallel trial involves the evaluation of the hypothesized beneficial effect of adherence to a Mediterranean-style diet in subjects at high cardiovascular risk on surrogate vascular markers encompassing vascular health indices such as endothelial function indices and ceramide plasma pathways. Such markers could be regulated by amelioration of the lipidaemic profile and by modulation of markers of inflammatory adipose dysfunction such as adipokine serum levels.

## Aims

- **The first aim was to evaluate** the effects of adherence to a Mediterranean-style diet on some vascular health indices, such as endothelial function and arterial stiffness markers at the 6-(T1) and 12-month (T2) follow-ups.

- **The second aim was to evaluate t**he effects of adherence to a Mediterranean-style diet on the lipidaemic profile and on some serum ceramide levels at the 6-(T1) and 12-month (T2) follow-ups.

- **The third aim was to** analyse the effects induced by adherence to a Mediterranean-style dietary regimen on the modulation of serum concentrations of some adipokine serum levels at the 6-(T1) and 12-month (T2) follow-ups.

## Materials and methods

### Patients and recruitment

We recruited all consecutive patients at high risk of cardiovascular diseases admitted to the Internal Medicine and Stroke Care ward at the University Hospital of Palermo, Italy between September 2017 and December 2020.

### Enrolment criteria:

The enrolled patients were male subjects aged between 55 and 80 and female subjects between the ages of 60 and 80 who tested positive for at least two of the following eligibility criteria thus to be classified as subjects at high cardiovascular risk

1. Type 2 diabetes mellitus

2. Arterial hypertension

3. Body mass index (BMI) $\geq$ 25

4. Active smoking

5. Family history of early cardiovascular disease

6. Previous cardiovascular or cerebrovascular events ($>$ 6 months)

### Exclusion criteria

All patients with recent ($<$6 months) cardiovascular or cerebrovascular events were excluded

### Definition of risk factors

We collected a careful medical history for each patient at the time of hospitalisation, and performed a detailed physical examination. Each subject received routine laboratory tests and M-mode and 2D echocardiogram.

Diabetes was defined according to the ADA 2021 criteria or by a previous dietary, oral anti-diabetic or insulin treatment before hospital admission [17].

Arterial hypertension was assessed using the ESC/ESH 2018 criteria [18].

Hypercholesterolemia was defined as serum total cholesterol levels $\geq$ 5.1 mmol/L, while hypertriglyceridemia was defined as values $\geq$ 1.7 mmol/L following NCEP-ATP III criteria [19].

The study protocol was approved by the institutional review board, and the study was performed in accordance with the principles of the Declaration of Helsinki and Good Clinical Practice [20].

Each patient provided written informed consent before participation.

The trial has been registered in Clinical Trials.gov [ClinicalTrials.gov Identifier: NCT04873167] on 05/05/2021 and it has been designed and conducted according criteria pre-established by CONSORT (*Consolidated Standards of Reporting Trials*) (see Fig 1).

**Ethics approval and consent to participate.** This protocol study was approved by the Ethics Committee of the Policlinico P. Giaccone Hospital, and all patients provided written informed consent to participate in the study and for sampling and banking of the biological material. Written informed consent was obtained from all the patients. A statement of ethics approval was obtained using the name of the ethics committee and reference number if appropriate

**Consent for publication.** All enrolled patients (or legal parents or guardians for children) provided consent to publish individual patient data.

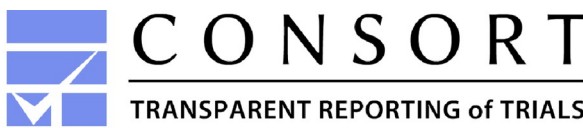

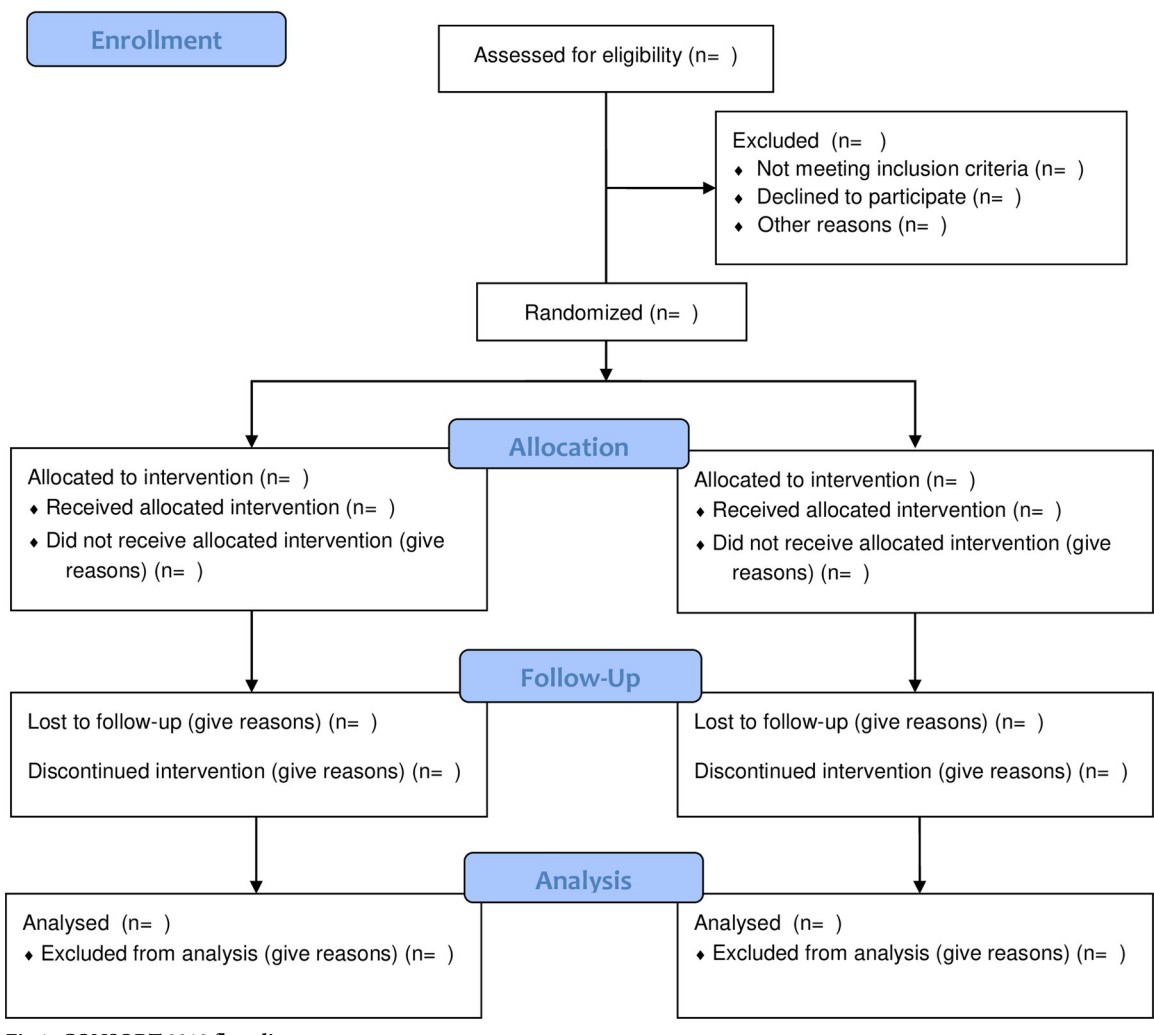

**Fig 1. CONSORT 2010 flow diagram.**

## Randomization

The enrolled subjects, after the evaluation of the degree of adherence to a dietary regimen of the Mediterranean-style diet (5), were randomised to two different types of dietary schemes:

1. Group A (experimental arm) - Mediterranean Diet: adherence to a Mediterranean-style diet was assessed through dietary screening at each follow up visit (every three months) for the entire duration of the study (twelve months).

2. Group B (control arm) - Low-fat diet: for which the enrolled subjects received dietary "counselling" starting from their first visit at the time of enrolment and, subsequently, every three months for the entire duration of the study (twelve months).

The randomization code was based on computer-generated random numbers and a 2:1 randomization ratio for experimental/control arm was performed both to contain the high cost of clinical investigations to be performed, and to avoid the effects of a high dropout rate by allowing more power for a per-protocol analysis. As a consequence of the type of randomization chosen, in order to maintain the power of the study, an increase in sample size of 12% was planned.

The investigators that collected clinical data of enrolled patients are not the same investigators involved in patient treatment. They also were not involved in the randomization process.

## Endpoints of the study

1. Effects of adherence to Mediterranean Diet on some surrogate markers of vascular damage, such as endothelial function measured by means of the reactive hyperaemia index (RHI) and augmentation index *(AIX)*, at the 6- and 12-month follow-ups

2. Effects of adherence to Mediterranean Diet on the lipidaemic profile *(serum total cholesterol, serum LDL cholesterol, serum triglycerides)* at the 6-month (T1) and 12-month (T2) follow-ups

3. Effects of adherence to Mediterranean Diet on serum levels of ceramides at the 6- and 12-month follow-ups.

4. Effects of adherence to Mediterranean Diet on serum levels of visfatin, adiponectin and resistin at the 6- and 12-month follow-ups.

**Dietary intervention trial.** We adopted the dietary intervention protocol with regard of Mediterranean Diet and low-fat diet validated by the PREDIMED trial [5].

*A. Mediterranean Diet***:** The general guidelines to follow the Mediterranean diet that researchers provided to participants included the following positive recommendations a) abundant use of olive oil for cooking and dressing dishes; b) consumption of ≥ 2 daily servings of vegetables (at least one of them as fresh vegetables in a salad), discounting side dishes; c) ≥ 2–3 daily servings of fresh fruits (including natural juices); d) ≥ 3 weekly servings of legumes; e) ≥ 3 weekly servings of fish or seafood (at least one serving of fatty fish); f) ≥ 1 weekly serving of nuts or seeds; g) select white meats (poultry without skin or rabbit) instead of red meats or processed meats (burgers, sausages); h) cook regularly (at least twice a week) with tomato, garlic and onion adding or not other aromatic herbs, and dress vegetables, pasta, rice and other dishes with tomato, garlic and onion adding or not aromatic herbs. This sauce is made by slowly simmering the minced ingredients with abundant olive oil. Negative recommendations are also given to eliminate or limit the consumption of cream, butter, margarine, cold meat, pate, duck, carbonated and/or sugared beverages, pastries, industrial bakery products (such as cakes, donuts, or cookies), industrial desserts (puddings, custard), french fries or potato chips, and out-of-home pre-cooked cakes and sweets. The researchers insisted that two main meals per day should be eaten (seated at a table, lasting more than 20 minutes). For usual drinkers, the advice was to use wine as the main source of alcohol (maximum 300 ml, 1–3 glasses of wine per day). If wine intake was customary, a recommendation to drink a glass of wine per day (bigger for men, 150 ml, than for women, 100 ml) during meals was given. *Ad*

*libitum* consumption was allowed for the following food items: nuts (raw and unsalted), eggs, fish (recommended for daily intake), seafood, low-fat cheese, chocolate (only dark chocolate, with more than 50% cocoa), and whole-grain cereals.

**B. Control diet group.** A low-fat, high complex carbohydrate diet, as recommended by the National Cholesterol Education Program, with <30% of total calories from fat (12–14% MUFAs, 6–8% PUFAs, < 10% SFAs), 55% from carbohydrates and 15% from protein. In both diets, the cholesterol content was adjusted to<300 mg/day. The focus in the control group was to reduce all types of fat, with particular emphasis in recommending the consumption of lean meats, low-fat dairy products, cereals, potatoes, pasta, rice, fruits and vegetables.

## Evaluation of adherence to the Mediterranean and to the low-fat diet

**As indicated in previous research studies conducted by our own group** [21–23]**, for** participants in the Mediterranean-diet group, researcher held individual and group dietary-training sessions at the baseline visit and every three months thereafter. In each session, participants completed a 14-item dietary questionnaire to assess adherence to the Mediterranean diet so that personalized advice could be provided to the study participants in these groups.

Information relating to the dietary habits of the enrolled patients was collected through a specific questionnaire (semiquantitative food frequency questionnaire (F.F.Q.s) [24] dedicated to the evaluation of the frequency of different food items in the diet and adapted to the Italian population. The subjects enrolled were classified based on the levels of adherence to a Mediterranean-style diet according to the methods proposed by Trichopoulou et al. [25].

The patients enrolled in the study were assessed for their adherence to a dietary regimen of the Mediterranean-style diet using the Mediterranean Diet Score, and the quantity and frequency of consumption of the food items characterising a Mediterranean-style diet were evaluated.

The consumption of foods presumed to be far from this diet pattern (i.e., rare or monthly consumption, meat and meat products, poultry and complete fat products) were assigned scores on an inverse scale.

A value of 0 or 1 was assigned to each indicated food component. For the beneficial components, patients whose consumption was below the median were assigned a value of 0. For components presumed to be detrimental, patients whose consumption was below the median were assigned a value of 1.

We have applied a non-monotonous function for the alcohol, i.e., score 5 for the consumption of less than 300 ml of alcohol per day, score 0 for no consumption or for the consumption of > 700 ml per day and scores from 4 to 1 for the consumption of 600–700, 500–600, 400–500 and 300–400 ml per day (100 ml is equivalent of 12 g ethanol).

Participants in the control group also received dietary training at the baseline visit and completed the 10-item questionnaire at baseline to assess their adherence to the Mediterranean diet. During admission visit of the study, they received a leaflet explaining the low-fat diet on three month basis and assessing adherence to this low-fat diet with the use of a separate 9-item dietary questionnaire. Scores ranged from 0 to 9, with higher scores indicating greater adherence to a low-fat diet.

## Biochemical analysis

Blood samples were obtained in the morning after at least ten fasting hours and were stored by cryopreservation at -80˚C [26].

Fasting plasma glucose levels, total cholesterol, triglycerides, HDL cholesterol, and LDL cholesterol were measured using traditional enzymatic methods at enrolment and at the 6- and 12-month follow-ups.

After 10 min of rest in the supine position, vital signs were recorded, and blood samples were collected from the antecubital vein. EDTA-anticoagulated peripheral blood was drawn from each patient within 12 h of symptom onset. Serum and plasma were immediately separated by centrifugation and stored in aliquots at -80˚C until analysis [26].

Adiponectin, resistin and IL-6 serum levels were measured by enzyme-linked immunosorbent assay (ELISA) according to the manufacturer's instructions. For adiponectin and resistin evaluation, high sensitivity kits (Biovendor) were used; IL-6 was determined by the Diaclone ELISA kit. Regarding the sensitivity of the adiponectin test (Biovendor), the analytical limit of detection was 0.6 microgram/ml; the intra- and interassay coefficients of variation (%) were 4.1 and 4.0, respectively. For the resistin assay (Biovendor), the analytical limit of detection was 0.1 ng/ml. Visfatin was measured by Sandwich ELISA (visfatin Phoenix Pharmaceuticals Inc.); the minimum detectable concentration for visfatin was 1.8 ng/ml.

## Endothelial function evaluation

The principle of RH-PAT [27] has been described previously. During the physical examination, a blood pressure cuff was placed on the upper limb, and the contralateral arm was used as a control. The pulse amplitude tonometry (PAT) probe was placed on one finger of each of the two hands. After 5 minutes of control measurement, the pressure cuff was inflated to 200 mmHg for 5 minutes to induce reactive hyperaemia and then deflated.

The RH-PAT data were analysed at enrolment and at the 6- and 12-month follow-ups. Measurement was made digitally using the Endo-PAT2000 software version 3.0.4 device. The RH-PAT index reflects the extent of reactive hyperaemia. It was calculated as the ratio of the average of the PAT signal amplitudes above the first minute of initial measurement to 1.5 minutes of measurement following deflation of the cuff (A: Control Arm; C: Occluded Arm) divided by the average of the PAT signal amplitudes over 2.5 minutes before inflation of the pressure cuff (B: Control Arm; D: Occluded Arm). This RH-PAT index, called the reactive hyperaemia index (RHI), is expressed by the formula RHI = (C/D)/(A/B) x basal correction.

**Study samples and ceramide dosage.** All analyses used fasting (fasting for ≥8 hours) plasma EDTA samples collected at baseline and at the follow-up visits. All samples were processed no later than 2 hours after collection and stored in −80˚C freezers. Samples from experimental and control groups were randomly distributed before being shipped to the laboratory for metabolomics assays. LC-MS techniques were used to quantitatively profile ceramides in plasma samples [27,28]. Circulating levels of chosen ceramides (Cer), ceramide 16:0 (Cer-16), ceramide 18.0 (Cer-18), ceramide 24 (Cer-24), ceramide 22 (Cer-22), ceramide 24/ceramide18 (Cer-24/Cer-18) ratio in human plasma were determined by liquid chromatography coupled to a tandem mass spectrometry (LC–MS/MS) system, as previously described with slight modifications. Briefly, 10 μL of plasma sample was spiked with 100 μL of methanolic ice-cold internal standard solution containing a mixture of 11 deuterated compounds supplied by Avanti Polar Lipids. An additional 100 μL of ice-cold methanolic solution was added. After vortexing and centrifugation (5 min, 3500 rpm, 4˚C), the supernatant was transferred to an HPLC vial and 5 μL was injected into the LC-MS/MS system [28,29].

The chromatographic separation of the lipid species was performed using an Acquity UPLC instrument (Waters Associates, Milford, MA, USA) operated using the MassLynx 4.1 software. The monitoring and quantification of the lipids was performed in the MRM mode using two different 5-min acquisition methods. The concentrations of ceramides were

calculated by using external calibration curves with authentic standards [28,29]. Plasma ceramide concentrations (μg/mL) were assayed using a validated LC-MS/MS protocol at enrolment and at the 6- and 12-month follow-ups.

## Statistical analysis

A sample size (performed by IBM SPSS SamplePower 3) of 45 patients per group would have 80% power to detect a difference in means of RHI values of 15%, assuming that the common standard deviation is 0.5, using a two-group *t*-test with α = 0.05 two-sided significance level. Because of the type of randomization chosen (2:1 ratio), in order to maintain the power of the study, an increase in sample size of 12% was planned. A total of 101 patients were randomized to the Mediterranean diet arm and 52 to the low-fat diet arm.

Statistical analysis of quantitative and qualitative data, including descriptive statistics, was performed for all items. The normal distribution of continuous variables was assessed by Kolmogorov–Smirnov test, and since the variables have been found to be normally distributed, they have been presented as mean and standard deviation (SD). Categorical variables have been summarized as absolute number (percentage).

Frequency analysis was performed using the Pearson's chi-square test and Fisher exact test, as needed. In multiple comparisons the significant values were corrected using the Bonferroni correction. The one-way analysis of variance (ANOVA) was used to compare parametric variables between the different treatment groups. The one-way repeated measures ANOVA test was used to compare the different variables at the different intervals for each study group.

Multivariable regression analysis was used to examine the correlations between some clinical patient parameters (dependent variables), as vascular health indices (RHI and AIX), lipidaemic profile and some adipokine serum levels, and Mediterranean diet effect (independent variable) at the different follow up (0, 6 and 12 months) in a multiple regression model.

Data were analyzed by IBM SPSS Software 24 version (IBM Corp., Armonk, NY, USA). All p-values were two-sided and p≤0.05 was considered statistically significant.

## Results

We enrolled 101 patients randomised to the Mediterranean Diet and 52 control subjects randomised to a low-fat diet with a dietary "counselling" between September 2017 and December 2020. No patient was lost at follow-up in both the groups.

Clinical, demographic, and laboratory variables of patients at the time of enrolment are shown in Table 1.

The univariate analysis showed that subjects in the Mediterranean Diet group showed significantly lower mean creatinine values (1.1 ± 0.4 vs. 1.4 ± 1.2, p = 0.010), significantly higher mean CRP values (11.1 ± 10.8 vs. 5.6 ± 7.3, p = 0.002) and significantly higher ejection fraction (EF) mean values (48.8 ± 8.8 vs. 27.1 ± 26.3, p <0.0005) than those in the control group. With regard to biochemical variables, mean serum levels of resistin was higher among subjects in the Mediterranean Diet group than those in the control group (10.8 ± 1.5 vs. 10.2 ± 1.6, p = 0.037), as well as the values of Cer -24 (2.23 ± 0.19 vs. 2.15 ± 0.21, p = 0.030), Cer-16 (0.15 ± 0.01 vs. 0.14 ± 0.01, p <0.0005) and Cer-22 (0.60 ± 0.03 vs. 0.55 ± 0.04, p <0.0005), with statistically significant inter-group differences. There were also more smokers and patients with chronic kidney disease and carotid plaque among the subjects in the Mediterranean Diet group and the difference was statistically significant.

Table 2 shows the quantitative and qualitative variables of the two groups at baseline, at the six-month (T1) and at the twelve-month (T2) follow-up.

**Table 1. Clinical and laboratory variables at baseline in subjects treated with Mediterranean diet and in control group.**

| VARIABLES | MEDITERRANEAN DIET GROUP (N = 101) | LOW-FAT GROUP (N = 52) | P Inter Group |
|---|---|---|---|
| Age (years) (mean±SD) | 68 ± 10.6 | 61 ± 7.7 | 0.055 |
| Sex (M/F) N | 49/49 | 34/17 | 0.13 |
| SBP (mmHg) (mean±SD) | 128 ± 10.8 | 127 ± 14.3 | 0.550 |
| DBP (mmHg) (mean±SD) | 78 ± 12.3 | 75 ± 13.1 | 0.122 |
| Weight (kg) (mean±SD) | 85 ± 17.5 | 83 ± 12.7 | 0.354 |
| Height (cm) (mean±SD) | 167 ± 5.3 | 166 ± 6.7 | 0.160 |
| BMI (kg/m$^2$) (mean±SD) | 28.7 ± 5.6 | 30.2 ± 4.9 | 0.104 |
| Microalbuminuria (mg/24h) (mean±SD) | 141 ± 97 | 185 ± 96 | 0.992 |
| Creatinine (mg/dL) (mean±SD) | 1.1 ± 0.4 | 1.4 ± 1.2 | **0.010** |
| EGFR (ml/min) (mean±SD) | 68 ± 26.9 | 64 ± 47.9 | 0.480 |
| ESR (mm/h) (mean±SD) | 18.9 ± 16.4 | 13.9 ± 11.8 | 0.138 |
| CRP (mg/dL) (mean±SD) | 11.1 ± 10.8 | 5.6 ± 7.3 | **0.002** |
| EF (%) (mean±SD) | 48.8 ± 8.8 | 27.1 ± 26.3 | **< 0.0005** |
| RWT N (%) | 62 (61) | 35 (67) | 0.295 |
| LAV N (%) | 68 (67) | 25 (48) | **0.029** |
| LVMI N (%) | 46 (45) | 14 (26) | 0.084 |
| Ischemic heart disease N (%) | 39 (38) | 25 (48) | 0.171 |
| Diabetes N (%) | 55 (54) | 29 (55) | 0.507 |
| PAD N (%) | 12 (11) | 5 (9) | 0.44 |
| Carotid plaque N (%) | 95 (94) | 38 (73) | **< 0.0005** |
| Stroke /TIA N (%) | 14 (13) | 5 (9) | 0.316 |
| NFKDOQI stage N (%) | | | **< 0.0005** |
| I | 26 (25) | 16 (30) | |
| II | 32 (31) | 14 (26) | |
| IIIA/IIIB | 20 (19) / 17 (16) | 5 (4) / 3 (5) | |
| IV | 6 (5) | 0 | |
| V | 0 | 14 (26) | |
| Smoke n (%) | 40 (39) | 29 (55) | **0.042** |
| Statins (n/%) | 53 (52.47) | 25 (49.01) | 0.614 |
| Insulin (n/%) | 21 (20.79) | 6 (11.76) | 0.187 |
| basal | 18 (17.78) | 2 (3.91) | **0.020** |
| rapid | | | |
| Anti SGTT2 (n/%) | 41 (40.59) | 22 (43.13) | 0.862 |
| GLPI-ra agonists (n/%) | 25 (24.75) | 3 (5.88) | **0.004** |
| glinides (n/%) | 19 (18.81) | 0 | **0.001** |
| Metformin (n/%) | 20 (19.80) | 15 (29.4) | 0.227 |
| Sulfonilureas (n/%) | 1 (0.9) | 6 (11.76) | **0.006** |
| Ace-inhbitors (n/%) | 32 (31.68) | 28 (54.90) | **0.008** |
| ARBs (n/%) | 58 (57.42) | 28 (54.90) | 0.672 |
| CCB (n/%) | 33 (32.67) | 16 (31.37) | 0.811 |
| Beta blockers (n/%) | 61 (60.39) | 23 (45.09) | 0.062 |
| Tiazidic diuretics (n/%) | 13 (12.87) | 18 (35.29) | **0.003** |
| Furosemide (n/%) | 71 (70.29) | 22 (43.13) | **0.001** |
| MRA (n/%) | 18 (17.82) | 6 (11.76) | 0.578 |

*SBP*: *Systolic blood pressure*; *DBP*: *Diastolic blood pressure,*, HDL: Hiogh-density lipoprotein; LDL: Low density lipoprotein; E*GFR*: Estimated glomerular filtration rate; **ESR**: Erotrocyte sedinentation rate, **CRP**: C-reactive protein, **EF**: Ejection Fraction; antidiabetic drugs:

*LVMI*: *Left ventricular mass index*, **LAV**: Left atrial volume, *RWT*: Relative wall thickness, **RHI**: Reactive hyperaemia index, **AIX**: Augmentatioon index, **C24:0**: C24:0 ceramide; **C16:0**: C16:0 ceramide; C22:0: C22:0 ceramide, **C18:0**: C18:0 ceramide, **C24:0/C16:0**: C24:0/C16:0 ratio.

At the six-month follow-up (T1), subjects in the Mediterranean Diet group showed lower mean serum total cholesterol levels (144 ± 31.9 mg/dl vs. 163 ± 45.9 mg/dl, p = 0.003), compared to the control group (see Table 2).

At the six-month follow-up in subjects in the Mediterranean Diet group, compared to the control group we observed a higher increase in RHI values (2.1 ± 0.4 vs. 1.7 ± 0.6, p <0.0005) (see Table 2).

With regard to biochemical variables, patients in the Mediterranean Diet group also showed lower serum levels of resistin at the six-month follow-up compared to the control group (9.7 ± 1.0 vs. 10.2 ± 1.1, p <0.0005), higher values of adiponectin (28.9 ± 2.9 µg/mL vs. 23.2 ± 4.3 µg/mL, p <0.0005), lower values of visfatin (12.3 ± 1.8 ng/ml vs. 15.1 ± 3.3 ng/ml; p<0.0005), lower values of Cer- 24 (1.96 ± 0.28 µg/mL vs. 2.19 ± 0.23 µg/mL, p <0.0005), higher values of Cer-22 (0.60 ± 0.39 µg/mL vs. 0.57 ± 0.45 µg/mL, p <0.0005) and higher values of Cer-24/Cer-16 ratio (12.11± 0.90 ug/mL vs 11.60 ± 1.13 ug/mL; p = 0.003), with

**Table 2. Metabolic and lipidic variables and vascular health indexes in Mediterranean Diet group and control group at baseline (T0), at 6 (T1) and 12 month (T2) follow-ups.**

| | Mediterranean Diet group | Low-fat Diet Group | P* | Mediterranean Diet group | Low-fat Diet Group | P* | Mediterranean Diet Group | Low-fat diet group | P* |
|---|---|---|---|---|---|---|---|---|---|
| | T0 | T0 | | T1 | T1 | | T2 | T2 | |
| Total cholesterol (mg/dL) (mean±SD) | 170±18 | 169±48 | 0.834 | 144±31.9** | 163±45.9** | **0.003** | 134±26.7** | 156±42.8** | **<0.0005** |
| LDL cholesterol (mg/dL) (mean±SD) | 69.2±25 | 70.1±27 | 0.909 | 68±28.3 | 64±24.3 | 0.673 | 63±23.5§ | 74±18.0 | **0.009** |
| HDL cholesterol (mg/dL) (mean±SD) | 40 ± 147 | 40±15 | 0.458 | 40±13.9 | 39±14.0 | 0.408 | 42±16.9 | 40±13.1 | 0.430 |
| Triglycerides (mg/dL) (mean±SD) | 112 ± 76.4 | 107±49 | 0.695 | 95±24.6^ | 101±43.2** | 0.286 | 89±20.1 | 94±34.8** | 0.352 |
| RHI (mean±SD) | 1.7 ± 0.5 | 1.6 ± 0.6 | 0.997 | 2.1±0.4 | 1.7±0.6 | **<0.0005** | 2.2±0.3 | 1.7±0.5 | **<0.0005** |
| AIX(mean±SD) | 6.7 ± 20.6 | 8.5 ± 11.2 | 0.590 | 12.2±93 | 17.3±35.4 | 0.235 | 12.9±1.6 | 13.2±2.5 | 0.922 |
| Visfatin (ng/ml) (mean±SD) | 15.6 ± 3.2 | 22.4 ± 4.2 | 0.297 | 12.3±1.8** | 15.1±3.3 | **<0.0005** | 10.2±1.4** | 14.8±3.8 | **<0.0005** |
| Resistin (ng/ml) (mean±SD) | 10.8 ± 1.5 | 10.2 ± 1.6 | **0.037** | 9.7±1.0** | 10.2±1.1 | **0.003** | 8.2±0.9** | 10.2±1.1 | **<0.0005** |
| Adiponectin (microg/ml) (mean±SD) | 23.1 ± 3.9 | 22.4 ± 4.2 | 0.355 | 28.9±2.9** | 23.2±4.3§ | **<0.0005** | 23.4±4.6 | 23.2±4.3§ | 0.790 |
| C24:0 (ug/mL) (mean±SD) | 2.23 ± 0.19 | 2.15 ± 0.21 | **0.030** | 1.96±0.28** | 2.19±0.23 | **<0.0005** | 1.7±0.33** | 2.7±2.41 | **<0.0005** |
| C16:0 (ug/mL) (mean±SD) | 0.15 ± 0.01 | 0.14 ± 0.01 | < 0.0005 | 0.15±0.10 | 0.15±0.01 | 0.820 | 0.14± 0.10 | 0.14±0.24 | 0.795 |
| C22:0 (ug/mL) (mean±SD) | 0.60 ± 0.03 | 0.55 ± 0.04 | < 0.0005 | 0.60±0.39 | 0.57±0.45** | **<0.0005** | 0.57 ± 0.06** | 0.57±0.05** | 0.893 |
| C18:0 (ug/mL) (mean±SD) | 0.53 ± 0.05 | 0.52 ± 0.01 | 0.210 | 0.53±0.44 | 0.53±0.01 | 0.900 | 0.32 ± 0.01** | 0.59±0.04 | **<0.0005** |
| C24:0/C16:0 (ug/mL) (mean±SD) | 11.47 0.84 | 11.38 ± 0.91 | 0.550 | 12.11±0.90** | 11.60±1.13 | **0.003** | 13.34 ± 1.11** | 11.62±0.76 | **<0.0005** |

**HDL:** High-density lipoprotein; **LDL:** Low-density lipoprotein; **EGFR:** Estimated glomerular filtration rate; **ESR**: Erythrocyte sedimentation rate. **CRP:** C-reactive protein. **RHI:** Reactive hyperaemia index. **AIX:** Augmentation index. **C24:0**: C24:0 ceramide; **C16:0**: C16:0 ceramide; **C22:0**: C22:0 ceramide. **C18:0:** C18:0 ceramide, **C24:0/C16:0**: C24:0/C16:0 ratio

*intergroup analysis at each follow-up time; intragroup analysis:

^ p<0.05

§ p<0.005

**p<0.0005 vs T0.

statistically significant inter-group differences (see Table 2). With regard of body weight the mean body weight at six-month follow up were in Mediterranean group vs control group was similar (73.4± 1.1 vs 73.1±0.9).

At the twelve-month follow-up (T2), subjects in the Mediterranean Diet group showed significantly lower serum total cholesterol levels (134 ± 26.7 mg/dl vs. 156 ± 42.8 mg/dl, p<0.0005), lower serum LDL cholesterol (63.4 ± 23.5 mg/dl vs. 74 ± 18.0 mg/dl; p = 0.009), compared to the control group (see Table 2).

At the twelve-month follow-up, we observed a further significant increase in the mean RHI in the subjects of the Mediterranean Diet group (2.2 ± 0.3 vs. 1.7 ± 0.5; p<0.0005), whereas we observed no significant differences in the mean serum levels of adiponectin (23.4 ± 0.4 μg/mL vs. 23. 2 ± 0.4 μg/mL; p = 0.790), a more significant reduction of serum levels of resistin (8.2± 0.9 ng/ml vs. 10.2 ±1.1 ng/ml, p<0.0005) and visfatin (10.2 ± 1.4 ng/ml vs. 14.8 ± 3.8 ng/ml, p<0.0005); no significant change was observed with regard to AIX (see Table 2). With regard of body weight the mean body weight at six-month follow up in Mediterranean group vs control group was similar (73.2± 1.0 vs 73.2±0.9).

At the twelve-month follow-up, the same group also showed significantly lower values of Cer-24 (1.7 ± 0.33 μg/mL vs. 2.7 ± 2.41 μg/mL; p <0.0005), no significant difference in Cer-16 and Cer-22, significant lower serum levels of Cer-18 (0.32 ± 0.01 μg/mL vs. 0.59 ± 0.04 μg/mL p <0.0005) and higher values of Cer-24/Cer-16 ratio (13.34±1.11 ug/mL vs 11.62 ± 0.76 ug/mL; <0.0005) (see Table 2).

In subjects randomized to Mediterranean diet at intragroup analysis we observed a significant change of total cholesterol at T1 and at T2 vs T0 (p<0.0005) with a progressive lowering of serum total cholesterol across the follow-up visits. We also observed a significant intragroup change of LDL cholesterol mean serum levels at T2 vs T0 (p<0.005) with a further reduction of LDL serum level at 12 month follow-up (T2).

In Mediterranean group at intragroup analysis we also observed a progressive lowering of triglyceride serum levels at T1 vs T0 (p<0.05) and a progressive lowering of mean resistin serum levels and of visfatin mean serum levels at T1 and T2 vs T0 (p<0.0005 in both cases), whereas with regard of adiponectin mean serum levels we observed a significant increase of its mean serum levels at T1 vs T 0 (p<0.0005). With regard of ceramides, at intragroup analysis we observed a significant progressive reduction of mean serum levels of C24:0, C22:0, at T2 vs T0 (p<0.0005), and of mean serum levels of C18:0 at T2 vs T0 (p<0.0005), whereas with regard of C24:0/C16:0 we observed a significant increase of its mean serum levels at T1 and T2vs T 0 (p<0.0005).

In subjects randomized to low-fat diet at intragroup analysis we observed a significant change of total cholesterol at T1 and at T2 vs T0 (p<0.0005) with a progressive lowering of serum total cholesterol across the follow-up visits. In low-fat diet group at intragroup analysis we also observed a progressive lowering of triglyceride serum levels at T1and T2vs T0 (p<0.0005), whereas with regard of adiponectin we observed a significant increase of its mean serum levels at T1 and T2 vs T0 (p<0.005). With regard of ceramides, at intragroup analysis we observed a significant increase of mean serum levels of C:22 at T1 and T2 vs T0 (p<0.0005).

At multivariable regression analysis we observed a significant correlation between some clinical and lipidaemic variables and some adipokine and vascular health markers and a beneficial clinical effect of Mediterranean Diet on follow-up. In particular we observed a significant positive correlation between Mediterranean diet with RHI (Exp(B): 5,990; P = 0.0001), and Adiponectin (Exp(B): 1.472; p = 0.0001), and a significant negative correlation with AIX (Exp (B): 0.979; p = 0.032), Total cholesterol (Exp(B): 0.984; p = 0.002) visfatin (Exp(B): 0.675; p = 0.002), and resistin (Exp(B): 0.466; p = 0.005) at six month follow-up (see Table 3). On this

light, we also observed a significant positive correlation between Mediterranean diet with RHI (Exp(B): 14,420; P = 0.0001), and a significant negative correlation with AIX (Exp(B): 0.973; p = 0.034), Total cholesterol (Exp(B): 0.982; p = 0.001), LDL cholesterol (Exp(B): 0.983; p = 0.037) visfatin (Exp(B): 0.409; p<0.0005) and resistin (Exp(B): 0.091; p = 0.0001) at 12 month follow-up (see Table 3).

## Discussion

Our study revealed that adherence to a Mediterranean-style diet for an appropriate time significantly improved vascular health indices, including enhanced endothelial function, favorable alterations in lipid profiles and ceramide concentrations, and a reduction in inflammatory adipokines.

**Table 3. Multivariable regression analysis of the correlations between some clinical variables, vascular health indices, lipidaemic profile and adipokines, and Mediterranean diet effect (independent variable) at the different follow up (0, 6 and 12 months).**

| groups[a] | B | SE | Wald | P | Exp(B) | 95% CI Exp(B) | |
|---|---|---|---|---|---|---|---|
| RHI | 0,305 | 0,333 | 0,840 | 0,360 | 1,357 | 0,706 | 2,608 |
| AIX | -0,008 | 0,010 | 0,564 | 0,453 | ,992 | 0,973 | 1,012 |
| Total cholesterol | 0,000 | 0,006 | 0,006 | 0,940 | 1,000 | 0,989 | 1,012 |
| HDL | 0,002 | 0,012 | 0,040 | 0,842 | 1,002 | 0,979 | 1,027 |
| LDL | -0,002 | 00,003 | 0,534 | 0,465 | ,998 | 0,991 | 1,004 |
| Triglycerides | 0,001 | ,003 | 0,211 | 0,646 | 1,001 | 0,996 | 1,007 |
| Visfatin | 0,092 | 0,059 | 2,412 | 0,120 | 1,096 | 0,976 | 1,231 |
| adiponectin | 0,059 | 0,047 | 1,570 | 0,210 | 1,060 | 0,967 | 1,162 |
| **Resistin** | **0,299** | **0,130** | **5,269** | **0,022** | **1,348** | **1,045** | **1,740** |
| **At six month follow-up** | | | | | | | |
| groups[a] | B | SE | Wald | P | Exp(B) | 95% CI Exp(B) | |
| **RHI** | **1,790** | **0,436** | **16,869** | **0,0001** | **5,990** | **2,549** | **14,072** |
| **AIX** | **-0,021** | **0,010** | **4,575** | **0,032** | **0,979** | **0,960** | **0,998** |
| **Total cholesterol** | **-0,016** | **0,005** | **9,385** | **0,002** | **0,984** | **0,975** | **0,994** |
| HDL | 0,003 | 0,013 | ,046 | 0,830 | 1,003 | 0,978 | 1,029 |
| LDL | 0,012 | 0,007 | 2,901 | 0,089 | 1,012 | 0,998 | 1,026 |
| Triglycerides | -0,001 | 0,006 | ,054 | 0,816 | 0,999 | 0,987 | 1,010 |
| **Visfatin** | **-0,393** | **0,124** | **10,017** | **0,002** | **0,675** | **0,529** | **,861** |
| **Adiponectin** | **0,387** | **0,076** | **25,632** | **0,000** | **1,472** | **1,268** | **1,710** |
| **Resistin** | **-0,763** | **0,269** | **8,030** | **0,005** | **0,466** | **0,275** | **,790** |
| **At twelve month follow-up** | | | | | | | |
| Groups[a] | B | SE | Wald | P | Exp(B) | 95% CI Exp(B) | |
| **RHI** | **2,669** | **0,511** | **27,246** | **0,000** | **14,420** | **5,294** | **39,276** |
| **AIX** | **-0,027** | **0,013** | **4,513** | **0,034** | **0,973** | **0,949** | **0,998** |
| **Total cholesterol** | **-0,018** | **,006** | **10,479** | **,001** | **0,982** | **0,971** | **0,993** |
| HDL | 0,004 | 00,012 | 00,091 | ,764 | 1,004 | 0,980 | 1,028 |
| **LDL** | **-0,017** | **0,008** | **4,365** | **0,037** | **0,983** | **0,967** | **0,999** |
| Triglycerides | 0,000 | 0,007 | 0,005 | 0,944 | 1,000 | 0,986 | 1,014 |
| **Visfatin** | **-0,893** | **0,243** | **13,487** | **0,000** | **,409** | **0,254** | **0,659** |
| Adiponectin | 0,101 | 0,112 | 0,820 | 0,365 | 1,107 | 0,889 | 1,378 |
| **Resistin** | **-2,401** | **0,551** | **18,960** | **0,000** | **0,091** | **0,031** | **0,267** |

[a]: Reference: Control group.

**HDL:** High-density lipoprotein; **LDL:** Low-density lipoprotein; **RHI:** Reactive hyperaemia index. **AIX:** Augmentation index.

These findings support the diet's potential as a holistic approach to mitigate cardiovascular risk factors and underscore its role in promoting overall cardiometabolic health.

The first aim of our study was indeed to was to evaluate the effects of adherence to a Mediterranean-style diet on some vascular health indices, such as endothelial function and arterial stiffness markers. We reported that at the six-month and twelve-month follow-ups, subjects in the Mediterranean Diet group experienced a more significant increase in RHI values than those in the control group.

Reactive hyperaemia is a well-established technique for the non-invasive assessment of peripheral microvascular function and a predictor of all-cause and cardiovascular morbidity and mortality.

Hypoxia stimulates vasodilation through mechanisms involving prostacyclin ($PGI_2$)-, adenosine-, nitric oxide (NO)-, and $K^+$ channel-mediated hyperpolarization [27].

The reactive hyperaemia index (RHI) as measured by pulse amplitude tonometry (PAT), represents a simple, non-invasive measurement of endothelial function. Lower RHI has been correlated with cardiovascular (CV) risk factors, including obesity, total/HDL cholesterol ratio, diabetes, smoking and dyslipidaemia.

Endothelial and arterial stiffness indices have been defined as "surrogate" cardiovascular risk markers. The endothelium has been reported as an autocrine-paracrine organ. It has a crucial role in regulating tone and vascular structure in the control of humoral, nervous and mechanical inducements.

Our findings indicate that the adherence to a Mediterranean-style diet can correct endothelial dysfunction and increase mean RHI values.

A previous study [16] showed similar findings in patients at high cardiovascular risk. In the CORDIOPREV 805, patients with completed endothelial function study at baseline were randomized to follow a Mediterranean diet with endothelial function measurement repeated after 1 year. Patients who followed the Mediterranean diet had higher mean values of flow-mediated dilation (FMD) compared with those in the low-fat diet, even in those patients with severe endothelial dysfunction. These results suggest that the Mediterranean diet better modulates endothelial function compared with a low-fat diet and is associated with a better balance of vascular homeostasis in high cardiovascular risk and severe endothelial dysfunction.

The Mediterranean-style diet is rich in legumes, nuts, unrefined grains and fish, all significant L-arginine sources that sustains NO production by endothelium [30].

Multiple pathological conditions, such as hypertension, congestive heart failure, hypercholesterolemia, diabetes and metabolic syndrome are associated with the reduced bioavailability of NO [31–33]. Therefore, an increase in NO bioavailability could reduce the severity of these diseases [34–37].

Adherence to the Mediterranean-style diet has also been report to be able to have effects on the inflammatory state. The Mediterranean-style diet in comparison to a western meat diet is associated with a lower serum concentration of inflammatory markers [38].

The MOLI-SANI study [39] suggested that adherence to a Mediterranean-style diet reduces oxidative stress [40], vascular inflammation [39] and endothelial dysfunction [41]. Furthermore, other authors reported that lower serum concentration of VCAM-1, ICAM-1, E-selectin and P-selectin, CRP and IL-6 and a downregulation of adhesion molecules expressed on the surfaces of lymphocytes and monocytes after three months of implementation of a Mediterranean-style diet [42].

**The second aim of our study was to analyse the e**ffects of adherence to a Mediterranean-style diet on the lipidaemic profile in subjects at high cardiovascular risk. At the 6- and 12-month follow-ups, subjects treated with a Mediterranean-style diet showed lower mean serum total cholesterol levels, and lower mean serum LDL cholesterol than the control group.

Some studies have reported that moderate or high adherence to the Mediterranean-style diet can protect against an increased risk of atherosclerotic plaque formation and may provide protection against clinical vascular events.

Our study confirmed previous studies showing a favourable effect of adherence to a Mediterranean-style diet on the blood lipid profile [15,16,43].

The results of the study confirmed the impact of dietary factors on blood lipids. They also provided additional evidence on the response of high-density lipoprotein cholesterol to diet in free-living populations.

In our study, lipidaemic effects were observed at the 6-month follow-up and furtherly confirmed at the twelve-month follow-up.

Our study also aimed to evaluate the serum concentrations of the following ceramides: Cer 16, Cer-18, Cer-22 and Cer-24.

At 6 months, we report in the subjects of the Mediterranean Diet group lower values of Cer-24 and higher values of the Cer-22 and Cer-24/ Cer-16 ratio with statistically significant inter-group variability.

The same study group at the twelve-month follow-up also showed significantly lower values of C24:0 and C18, and higher values of the Cer-24/Cer-16 ratio.

Ceramides are bioactive lipids involved in cellular responses and apoptosis. Ceramide and its metabolites have been reported as a converging point between overeating and metabolic abnormalities that increase the risk of cardiometabolic diseases, such as insulin resistance and inflammation.

The intervention with the Mediterranean-style diet therefore modified the damaging effect of higher ceramide concentrations on CVD risk. There are two potential mechanisms for the modulatory effects of this diet on the ceramide pathway. First, the consumption of relevant dietary components can directly affect ceramide biosynthesis.

A study by Mantovani et al. [44] for example, evaluated how the plasma concentrations of ceramides Cer-16, Cer-18, Cer-22and Cer-24 were associated with the presence of inducible myocardial ischaemia, thus being independent positive predictors of defects of myocardial perfusion. Wang et al. [30] evaluated the concentrations of the ceramides mentioned above in a cohort of patients in the PREDIMED study and found that a greater risk of experiencing major cardiovascular events correlated with their increase.

These results suggest a positive association between baseline plasma ceramide levels and cardiovascular events and indicate how adherence to a Mediterranean-style diet can influence the potential negative relationship between elevated plasma ceramide concentrations and CVD.

Furthermore, our findings show an increase in the Cer-24/Cer-16 ratio, which appears to be inversely related to the risk of cardiovascular events, suggesting the utility of this ratio as a new biomarker of cardiovascular risk [45].

In our current study, we report that a higher degree of adherence to a Mediterranean-style diet induces a significant reduction in ceramide concentrations at the six-month and twelve-month follow-ups due a decrease in synthesis. This diet promotes an increase in the Cer-24/ Cer-16 ratio, which has a protective role against the risk of developing cardiovascular diseases.

Therefore, it is possible to hypothesise that greater adherence to a healthy lifestyle recommended by the Mediterranean Diet can improve the cardio- and cerebrovascular risk profile, which can be indirectly assessed by monitoring the ceramide ratio described above.

An additional objective of our research project was to evaluate the serum concentrations of inflammatory adipokines, correlating their change over follow-up visits with differences in the level of adherence to a Mediterranean-style dietary regimen, which may be related to an improvement in endothelial function as measured by RHI.

Our results showed that in subjects with a Mediterranean Diet adherence profile, compared to the control group in which a low-fat diet was prescribed, there was a significant reduction in the serum concentrations of resistin and visfatin, but an increase in the plasma levels adiponectin at the six-month follow-up. At the twelve-month follow-up, we observed a further, more significant reduction of resistin and visfatin serum levels in the Mediterranean Diet group, but a reduction of adiponectin serum levels in comparison to the levels at T1.

Human resistin plays a crucial regulatory role in the inflammatory response [46]. Resistin enhances the production of pro-inflammatory cytokines such as IL-12, IL-6, TNF-α and monocyte chemoattractant protein-1 (MCP-1) in PBMCs, macrophages and hepatic stellate cells through nuclear factor-κB (NF-κB) [47–49]. Furthermore, resistin inhibits the protein that mediates endothelial nitric oxide synthesis through oxidative stress in human endothelial cells [47] promotes the formation of human macrophages, induces a prothrombotic phenotype in human endothelial cells [48–49] and induces platelet activation by increasing the expression of P-selectin [49]. These results suggest that human resistin may play an essential regulatory role in modulating interactions between endothelial cells, and pathogenesis and progression of atherosclerosis [48,50]. The literature findings are consistent with the results obtained from a study conducted by our research team [26] reporting that diabetic subjects with a diabetic foot had higher plasma levels of IL-6 and resistin, defined as possible determinants of increased cardiovascular risk, but lower plasma levels of adiponectin.

Visfatin is one of the main adipokines secreted by adipose tissue. The visfatin level increases significantly in people with obesity due to increased BMI. Visfatin regulates immune cell trafficking and induces a chronic inflammatory state in adipocytes. It also induces insulin resistance [51]. Visfatin induces the formation of atherosclerotic regulating the extracellular matrix and angiogenesis degradation [51].

Current knowledge in this regard therefore suggests that an increase in visfatin and resistin levels contributes to the pathogenesis of endothelial dysfunction, which subsequently leads to a reduction in plasma concentrations of adiponectin [51]. This adipokine generally exerts an insulin-sensitizing, anti-inflammatory, and anti-apoptotic effect on many cell types and acts in the brain by increasing energy expenditure and inducing weight loss [52].

Considering the significant effect of the Mediterranean-style diet in reducing the levels of the adipokines mentioned above and increasing the synthesis of NO and adiponectin, it is likely that adherence to this dietary regimen can reduce the extent of endothelial dysfunction and, at the same time, keep the cardiovascular risk of these patients low.

These beneficial effects with regard of endothelial function evaluated by RHI and ISX and lipidaemic and adipokine effects have been also sustained by the results of multivariable regression analysis concerning the beneficial effects of education to the adherence to a Mediterranean Diet style.

The findings of this prospective study concerning the observed positive effects of adherence to a Mediterranean-style diet on adipokine serum levels, ceramide serum levels and an index of vascular health, such as RHI, may help to explain the cardiovascular and cerebrovascular protective relationship reported in previous studies from our own group with regard to ischaemic stroke, in particular, the atherosclerotic subtype [23] hemorrhagic cerebrovascular disorders[22] and congestive heart failure [21,53].

## Limitations

A possible limitation of the study could be represented by the finding of significantly higher mean serum CRP values in patients randomised to the Mediterranean Diet group than in

those of the control group. Nevertheless, this finding could be related to the higher frequency of smokers and patients with chronic kidney disease in the Mediterranean Diet group.

Additional limitations could be related to possible adherence bias to the diet style to which the patient was randomised to.

Furthermore, it would be interesting to evaluate cell adhesion molecules (CAMs) expression and endothelial progenitor cells (EPCs) frequency, although we evaluated reactive hyperemia index (RHI) that permit to avoid limitations of other methods of endothelial function evaluation such as flow-mediated dilation (FMD).

## Conclusions

Our findings reported the possible strict relationship between lipidaemic and beneficial vascular effects of Mediterranean diet underlying the pathogenetic interplay of some adipokines such as visfatin and resistin and of some ceramide serum levels as possible lipidaemic cardiovascular markers.

The findings of our current study offer a further possible explanation with regard to the beneficial effects of a higher degree of adherence to a Mediterranean-style diet on multiple cardiovascular risk factors and the underlying mechanisms of atherosclerosis. Moreover, these findings provide an additional plausible interpretation of the results from observational and cohort studies linking high adherence to a Mediterranean-style diet with lower total mortality and a decrease in cardiovascular events and cardiovascular mortality.

## Supporting information

**S1 File.**
(PDF)

## Author Contributions

**Conceptualization:** Mario Daidone, Antonino Tuttolomondo.

**Formal analysis:** Alessandra Casuccio.

**Investigation:** Maria Grazia Puleo, Alessandro Del Cuore, Gaetano Pacinella, Tiziana Di Chiara, Domenico Di Raimondo, Palmira Immordino.

**Methodology:** Alessandra Casuccio.

**Supervision:** Alessandra Casuccio, Antonino Tuttolomondo.

**Validation:** Alessandra Casuccio.

**Writing – original draft:** Mario Daidone.

**Writing – review & editing:** Mario Daidone, Antonino Tuttolomondo.

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
