## [Decision Letter · Decision Letter 0]

24 Jan 2024

PONE-D-23-37552Mediterranean diet effects on vascular health and serum levels of adipokines and ceramides.PLOS ONE

Dear Dr. Tuttolomondo,

Thank you for submitting your manuscript to PLOS ONE. After careful consideration, we feel that it has merit but does not fully meet PLOS ONE’s publication criteria as it currently stands. Therefore, we invite you to submit a revised version of the manuscript that addresses the points raised during the review process.

We look forward to receiving your revised manuscript.

Kind regards,

Aleksandra Klisic

Academic Editor

PLOS ONE

Journal Requirements:

5. Please include a copy of Table 2 which you refer to in your text on page 16.

Reviewers' comments:

Reviewer's Responses to Questions

**Comments to the Author**

1. Is the manuscript technically sound, and do the data support the conclusions?

Reviewer #1: Partly

Reviewer #2: No

Reviewer #3: Yes

2. Has the statistical analysis been performed appropriately and rigorously? 

Reviewer #1: No

Reviewer #2: Yes

Reviewer #3: Yes

3. Have the authors made all data underlying the findings in their manuscript fully available?

Reviewer #1: Yes

Reviewer #2: Yes

Reviewer #3: No

4. Is the manuscript presented in an intelligible fashion and written in standard English?

Reviewer #1: Yes

Reviewer #2: No

Reviewer #3: Yes

5. Review Comments to the Author

Reviewer #1: The manuscript addresses an interesting topic. The collected data are unique, the employed statistical methods rather basic and there is room left for improvements. My main comments are indeed on the performed statistical analysis.

1. It is nice to see that data are attached. Please, provide the code use to obtain the results in the tables as well.

Nevertheless, I feel unacceptable that the data are not anonymized. All patients information, also phone numbers, are available to anyone with access to the link in the paper. This is a clear violation of the privacy and it is rather sloppy and annoying.

2. The use of parametric tests is in general sound, but all tests are based on quite strong assumptions. In the main text, the authors state that they check for these assumptions, please provide evidence that they are met.

3. The employed statistical methods are too basic. I strongly suggest to consider random effects regression models to better account for the longitudinal structure of the data. As before, a residual analysis should be performed and discussed to ensure the reliability of the results. Instead of looking at one variable at the time, a full model should be considered and variable selection procedure implemented and discussed. I further suggest to discuss the potential impact of missing values, which are likely to arise in studies of this type (and it is not clear if any missing values is present in the current study too).

Reviewer #2: The study titled "Mediterranean diet effects on vascular health and serum levels of adipokines and ceramides" conducted by Daidone and colleagues evaluated the effect of Mediterranean diet on serum levels of adipokines and ceramides. The study is not well-designed. The topic is not novel and the way of presentation of results are not accurate. The English language needs extensive revisions by a native.

Reviewer #3: Daidone et al. have performed a study on the association of the Mediterranean diet with vascular health and serum adipokines. The study is well-conducted and well-written. I have some comments to consider:

- The introduction section is unnecessarily lengthy. The authors should try to focus on the main ideas and the rationale for performing the study.

- Figure 1 is not completed as the numbers are not added to the flowchart.

- The first paragraph of the discussion should focus on the main findings of the manuscript. It should be revised.

6. PLOS authors have the option to publish the peer review history of their article (what does this mean?). If published, this will include your full peer review and any attached files.

Reviewer #1: No

Reviewer #2: No

Reviewer #3: No

---

## [Author Response · Author response to Decision Letter 0]

22 Feb 2024

Dear Dr. Tuttolomondo,

Dear Editor thank you for the evaluation of our manuscript and for the chance to submit a revised version of our manuscript 

We ensured it

We upadated and corrected our data sharing 

We included a caption for each figure in our manuscript 

5. Please include a copy of Table 2 which you refer to in your text on page 16.

Table 2 is present on the text now 

We included the captions 

Reviewers' comments:

Dear Editor, 

Thank you very much for the useful revision comments and for the chance to submit a revised version of our manuscript.

You will find above our answers to the reviewer comments. 

5. Review Comments to the Author

Reviewer #1: The manuscript addresses an interesting topic. The collected data are unique, the employed statistical methods rather basic and there is room left for improvements. My main comments are indeed on the performed statistical analysis.

1. It is nice to see that data are attached. Please, provide the code use to obtain the results in the tables as well.

Nevertheless, I feel unacceptable that the data are not anonymized. All patients information, also phone numbers, are available to anyone with access to the link in the paper. This is a clear violation of the privacy and it is rather sloppy and annoying.

As indicated by the reviewer we removed the information available in the previous version of the attached data that have been enclosed only for a mistake. 

2. The use of parametric tests is in general sound, but all tests are based on quite strong assumptions. In the main text, the authors state that they check for these assumptions, please provide evidence that they are met.

Dear reviewer thanks for your interesting comments that could be important for the verification of the assumption of the observed significance. We have performed for all variables, in particular with regard of the quantitative variables, the test of normality of the data by kolmogorov-Smirnov test with Lilliefors correction. We found non-significant comforting results about the normality of the data and not relying only on the kolmogorov-Smirnov test. Thus, we repeated the data analysis also with nonparametric statistical procedures that provided identical results with the same significance emerged. Moreover, further sustainment concerning the results that have emerged can also be attributed to the fact that we compared patients at different study times through the paired data methodology, and this seemed to us to be further evidence of the goodness of fit of the data. Therefore, we have chosen to present the data only as means by reporting the parametric results according to ANOVA analysis.This choice is linked to our willness to not furtherly burden the text by reporting all the normality significances for the different variables.

3. The employed statistical methods are too basic. I strongly suggest to consider random effects regression models to better account for the longitudinal structure of the data. As before, a residual analysis should be performed and discussed to ensure the reliability of the results. Instead of looking at one variable at the time, a full model should be considered and variable selection procedure implemented and discussed. I further suggest to discuss the potential impact of missing values, which are likely to arise in studies of this type (and it is not clear if any missing values is present in the current study too).

We thank the reviewer again for this additional comment. Indeed in a previous draft of the manuscript we have had wondered of including some regression analyses that , also on the light of your suggestions, we have now decided to resubmit. It seemed to us that the results obtained and presented in the two tables were already sufficient to demonstrate the objectives of the study, but indeed the multivariable regression model for the variables seem to result adequate to be furtherly analyzed since lipid profile, markers of vascular damage such as RHI and AIX, and serum levels of visfatin, adiponectin, and resistin, could furtherly emphasizes the role of these clinical variables as markers of efficacy of the Mediterranean diet in the studied subjects. On the other hand, regarding the request for missing data, since this is a one-year study (6- and 12-month follow-ups) and not a large number of participants, we did not have a drop out of patients and the completeness of the collected data exceeds 98% for all variables. Consequently, we do not believe that this issue could have affected the results of our study.

Reviewer #2: The study titled "Mediterranean diet effects on vascular health and serum levels of adipokines and ceramides" conducted by Daidone and colleagues evaluated the effect of Mediterranean diet on serum levels of adipokines and ceramides. The study is not well-designed. The topic is not novel and the way of presentation of results are not accurate. The English language needs extensive revisions by a native

Dear reviewer 

Thank you for your comments

In our answers to reviewer 1 we added some further information on the statistical analysis and on the design of the study

With regard of the issue of novelty as expressed on the text the findings of this prospective study concerning the observed positive effects of adherence to a Mediterranean-style diet on adipokine serum levels, ceramide serum levels and an index of vascular health, such as RHI, may help to explain the cardiovascular and cerebrovascular protective relationship reported in previous studies from our own group with regard to ischaemic stroke, in particular, the atherosclerotic subtype (65), hamorrhagic cerebrovascular disorders (66) and congestive heart failure (67-69). This integrated evaluation is new, whereas the effects of a Mediterranean Diet on ceramide and adipokine serum levels it is novel finding. 

We furtherly revised the English style of the manuscript by a mother tongue revisor (see the Editing certification) 

Reviewer #3: Daidone et al. have performed a study on the association of the Mediterranean diet with vascular health and serum adipokines. The study is well-conducted and well-written. I have some comments to consider:

- The introduction section is unnecessarily lengthy. The authors should try to focus on the main ideas and the rationale for performing the study.

Dear reviewer thanks for your interesting comment. As suggested, we completely revised the introduction with the aim of making the paragraph more focused on the background that led us to design this study. 

- Figure 1 is not completed as the numbers are not added to the flowchart.

Dear reviewer, “see figure 1” has been written by mistake. We provided to delete it from the definitive manuscript 

- The first paragraph of the discussion should focus on the main findings of the manuscript. It should be revised.

Thanks again for your suggestion. We added a little paragraph in which are synthetizes the main findings of our study. 

6. PLOS authors have the option to publish the peer review history of their article (what does this mean?). If published, this will include your full peer review and any attached files.

Do you want your identity to be public for this peer review? For information about this choice, including consent withdrawal, please see our Privacy Policy.

Reviewer #1: No

Reviewer #2: No

Reviewer #3: No

---

## [Decision Letter · Decision Letter 1]

6 Mar 2024

Mediterranean diet effects on vascular health and serum levels of adipokines and ceramides.

PONE-D-23-37552R1

Dear Dr. Tuttolomondo,

We’re pleased to inform you that your manuscript has been judged scientifically suitable for publication and will be formally accepted for publication once it meets all outstanding technical requirements.

Kind regards,

Aleksandra Klisic

Academic Editor

PLOS ONE

Reviewers' comments:

Reviewer's Responses to Questions

**Comments to the Author**

1. If the authors have adequately addressed your comments raised in a previous round of review and you feel that this manuscript is now acceptable for publication, you may indicate that here to bypass the “Comments to the Author” section, enter your conflict of interest statement in the “Confidential to Editor” section, and submit your "Accept" recommendation.

Reviewer #1: All comments have been addressed

Reviewer #2: All comments have been addressed

Reviewer #3: All comments have been addressed

2. Is the manuscript technically sound, and do the data support the conclusions?

Reviewer #1: (No Response)

Reviewer #2: (No Response)

Reviewer #3: (No Response)

3. Has the statistical analysis been performed appropriately and rigorously? 

Reviewer #1: (No Response)

Reviewer #2: (No Response)

Reviewer #3: (No Response)

4. Have the authors made all data underlying the findings in their manuscript fully available?

Reviewer #1: (No Response)

Reviewer #2: (No Response)

Reviewer #3: (No Response)

5. Is the manuscript presented in an intelligible fashion and written in standard English?

Reviewer #1: (No Response)

Reviewer #2: (No Response)

Reviewer #3: (No Response)

6. Review Comments to the Author

Reviewer #1: (No Response)

Reviewer #2: (No Response)

Reviewer #3: (No Response)

7. PLOS authors have the option to publish the peer review history of their article (what does this mean?). If published, this will include your full peer review and any attached files.

Reviewer #1: No

Reviewer #2: No

Reviewer #3: No
